



# Multi-lidar wind resource mapping in complex terrain

Robert Menke[1], Nikola Vasiljević[1], Johannes Wagner[2], Steven P. Oncley[3], and Jakob Mann[1]

[1]Technical University of Denmark - DTU Wind Energy, Frederiksborgvej 399, 4000 Roskilde, Denmark
[2]Deutsches Zentrum für Luft- und Raumfahrt, Institut für Physik der Atmosphäre, 82234 Oberpfaffenhofen, Germany
[3]National Center for Atmospheric Research, Earth Observing Laboratory, Boulder, CO, USA

**Correspondence:** Robert Menke (rmen@dtu.dk), Jakob Mann (jmsq@dtu.dk)

**Abstract.** Scanning Doppler lidars have great potential for reducing uncertainty of wind resource estimation in complex terrain. Due to their scanning capabilities, they can measure at multiple locations over large areas. We demonstrate this ability using dual-Doppler lidar measurements of flow over two parallel ridges. The data have been collected using two pairs of long-range WindScanner systems operated in a dual-Doppler mode during the Perdigão 2017 measurement campaign. The lidars mapped the flow along the southwest and northeast ridges 80 m above ground level. By analyzing the collected data, we found that for different flow conditions on average wind speeds are 10% higher over the southwest ridge compared to the northeast ridge. At the southwest ridge, the data shows, depending on the atmospheric conditions, a change of 20% in wind speed along the ridge. For the measurement period, we have simulated the flow over the site using WRF-LES to compare how well the model can capture wind resources along the ridges. We used two model configurations. In the first configuration, surface drag is based purely on aerodynamic roughness whereas in the second configuration forest canopy drag is also considered. We found that simulated winds are underestimated in WRF-LES runs with forest drag due to an unrealistic forest distribution on the ridge tops. The correlation of simulated and observed winds is, however, improved when the forest parameterization is applied. WRF-LES results without forest drag overestimated the wind resources over the southwest and northeast ridges by 6.5% and 4.5% respectively. Overall, this study demonstrates the ability of scanning lidars to map wind resources in complex terrain.

## 1 Introduction

Traditionally, wind resource assessment is done with mast-mounted cup or sonic anemometers. Nowadays, with the commercialization and increasing acceptance of remote sensing devices such as lidars and sodars, this practice is changing due to clear advantages of remote sensing devices: they are easily deployed, cheaper, avoid the requirement of building permits and can measure at higher heights. However, mast based instruments, especially sonic anemometers, are probably still better suited for turbulence measurements (Sathe and Mann, 2013).

Vertically profiling wind lidars gained popularity for the assessment of mean wind speeds and are getting recognized by international standards for wind resource and power performance assessments (Clifton et al., 2018). Most profiling lidars perform velocity–azimuth display (VAD) scans to estimate the horizontal velocity from line-of-sight (LOS) measurements under the assumption of horizontal homogeneity. However, this assumption is typically violated in complex terrain. Errors from profiling lidars can be up to 10% when measuring in complex terrain as shown by Bingöl et al. (2009). One solution to





overcome this problem is to use several lidars that directly measure different components of the wind at the same location. Moreover, the deployment of several lidars with scanning capabilities allows the assessment of wind conditions over large areas (Vasiljević et al., 2019) which can give important insights into the spatial variability of flow over very complex terrain. Multi-lidars have been proven to have a high measurement accuracy in comparison studies with sonic anemometers (Pauscher

et al., 2016). Moreover, many studies utilized the scanning capability to measure wind fields over large areas assessing, for example, wind turbine wakes (Bingöl et al., 2010; Käsler et al., 2010; Trujillo et al., 2011; Iungo et al., 2013; Bodini et al., 2017; Menke et al., 2018b), the inflow towards wind turbines (Mikkelsen et al., 2013; Simley et al., 2016; Mann et al., 2018), the influence of surface and terrain features on the flow (Lange et al., 2016; Mann et al., 2017) and atmospheric phenomena such as gravity waves (Lehner et al., 2016; Palma et al., 2019).

In this publication, we will present measurements from the Perdigão 2017 campaign (Fernando et al., 2019). For this measurement campaign, wind lidars were a key measurement technology for the assessment of the flow over the complex terrain site. In total 7 profiling and 19 scanning lidars were deployed. The present study focuses on a subset of the entire data collection containing measurements of wind resources along the two parallel ridges.

The relevance of measurements is especially important for complex terrain where the uncertainty of current flow models

is high (Bechmann et al., 2011). Potential sources of error are the characterization of the roughness resulting from different types of canopies (Wagner et al., 2019a), the characterization of the stratification in the atmosphere (Palma et al., 2019), the description of the terrain (Lange et al., 2017; Berg et al., 2018) and model resolution which may not capture all important flow phenomena in complex terrain. We compare our measurements to a WRF-LES simulations with and without a parametrization of forest drag (Wagner et al., 2019a, b).

This paper is organized in the following way: Section 2 gives an overview of the Perdigão field campaign including a description of lidar and mast measurements, Section 3 presents the WRF model setup. Section 4 introduces the applied data processing techniques. The results and discussion of the data analysis are given in section 5, followed by our conclusions in section 6.

## 2 Field campaign overview

The Perdigão 2017 field campaign took place at a site centered at the village Vale do Cobrão located in Portugal close to the Spanish border. The main selection criteria for the site was a distinct terrain feature of two parallel ridges of 4 km in length (Figure 1). The ridges are about 1.5 km apart and the height difference from the valley bottom to the ridge tops is about 250 m. The northwest – southeast orientation of the ridges is perpendicular to the prevailing wind directions which were assessed previously to the campaign with a 30-m measurement mast (Vasiljević et al., 2017).

During the 2017 campaign, measurement devices were set up with a very high density by a large international group of universities, institutions and industry partners. Instruments were operated from early 2017 until early 2018 with an Intensive Operation Period (IOP) from May 1st to June 15th 2017. To map the flow over the measurement site 186 3-component sonic anemometers were installed on 50 meteorological masts with heights up to 100 m. Also 26 wind lidars (7 profiling lidars and 19



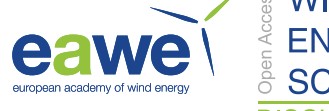

scanning lidars) were deployed. A full overview of the campaign's objectives and instrumentation may be found in Fernando et al. (2019). For this study, we analyze measurements from 4 long-range WindScanners lidars (Vasiljevic et al., 2016) and 4 measurement masts located on the ridge tops.

**Figure 1.** (a) Elevation map of the Perdigão site in the PT-TM06/ETRS89 coordinate system. (b) Tree height map. (c) View from the southwest of the ridges with lidar and sonic anemometer sampling positions, and wind turbine at center of the southwest ridge.

## 2.1 Lidar measurements

5   As mentioned above, for this study we analyze measurements of four out of the eight WindScanners that were operated by DTU during the measurement campaign. Specifically, we are focusing on a measurement scenario designed to measure the wind





resources along the two ridges which was achieved using the so-called ridge scans (Vasiljević et al., 2017). In the following sections we will describe the experiment layout design process, the deployment process including the calibration procedure, and the design and configuration of the scanning trajectories.

### 2.1.1 Layout

The layout of the 2017 experiment is an extension of the design introduced in the 2015 pilot Perdigão experiment (Vasilje-vić et al., 2017). In 2015, our focus was to measure wind resources above the southwest (SW) ridge using a single pair of WindScanners since the ridge tops are areas characterized with high wind resources and thus often used as locations for wind turbines placement in complex terrain. Also, current state-of-the-art flow models have difficulties predicting the flow behavior over the steep ridges. Therefore creating a good measurement dataset of the flow over such terrain is imperative to improve the

models.

In 2015 we established a traverse line of intersecting lidar beams about 2 km in length along the SW ridge 80 m above the ground level (AGL). This altitude was chosen to match the hub height of the wind turbine located on the SW ridge. With only a pair of lidars used to measure along each traverse line, it is not possible to resolve the vertical velocity component. Thus, the lidar positions and scan strategy needed to be chosen to keep the elevation angles of the laser beams as low as possible

(preferably below $5°$). Also, the intersecting angle between the laser beams must be at least $30°$. Having elevation angles below $5°$ensures that the influence of the vertical wind component is kept below 0.5% as as $\cos(5°) = 99.6$. In 2015, we placed a single pair of WindScanners on the northeast (NE) ridge, bringing them close the height of the traverse above the SW ridge to keep the elevation angles low. The separation of the WindScanners was about 1.2 km to match the design conditions for intersecting angles (Vasiljević et al., 2017). For the 2017 campaign, we extended our goal by also placing WindScanners on

the SW ridge, thus enabling simultaneous probing of wind resources above both ridges (Figure 1). The in-field placement of the lidars was done using high precision terrain data and orthophotos acquired prior to the Perdigão 2015 campaign (Vasiljević et al., 2017).

### 2.1.2 Deployment

The Perdigão site is rural, characterized by rugged terrain accessible with a limited off-road route network. Actually, the

network had to be extended to allow access to several instruments. In 2015, the WindScanners were powered by a diesel generator, whereas in 2017 a dedicated power grid was constructed to supply uninterrupted power to the devices. For these reasons, the initial deployment of instruments was complicated and time-consuming.

After the WindScanners were positioned at their designated locations, their orientation and leveling were determined by mapping the lightning rods of measurement masts using the WindScanners' laser beams (Vasiljevic, 2014, p.157). Both the

position of WindScanners and lighting rods had been measured with centimeter accuracy (Menke et al., 2019a). By comparing referenced and mapped positions the leveling and orientation of WindScanners were improved resulting in a pointing accuracy of about $0.05°$. To retain the pointing accuracy, the target mapping was repeated several times during the campaign to ensure that the leveling and orientation of the WindScanners remained unchanged.



### 2.1.3 Scanning strategy

The two traverses, which follow the ridge top line 80 m AGL, were designed using the high precision terrain data. The traverses were 1.8 km long described by points evenly spaced every 20 m. Accordingly we programmed the WindScanners to measure continuously along the traverses by moving the beams through the traverse points with the speed of $40\,\mathrm{m\,s^{-1}}$ and an accumu-
lation time of 500 ms. One scan took 48 s of which 45 s were spent on measurements, 0.5 s for acceleration and deceleration of the scanner heads and 2 s to return to the trajectory start point. Range gates were placed every 10 m, starting at 700 m, and extending to 2640 m (Table 1).

Typically, the WindScanner system uses a master computer to keep the synchronization of WindScanners to about 10 ms (Vasiljevic et al., 2016). This synchronization requires a stable network connection between the WindScanners and the master
computer. At the Perdigão site the systems were connected via directional long-range WiFi antennas which had tendency to have a low availability and/or high latency. Due to the unstable network the WindScanners were configured to start the measurements in a scheduled fashion according to GPS time, thus independently from the master computer. This introduced time offsets due to a system dependent startup time which varies over time and among the different WindScanners. However, the WindScanners could perform measurements independent of the network connection which results in higher data availability.
The average time offset between WS5 and WS6 is $0.42\pm1.03$ s and $0.7\pm0.65$ s between WS7 and WS8.

**Table 1.** WindScanner coordinates and details about the measurement settings.

| WindScanner | WS5 | WS6 | WS7 | WS8 |
|---|---|---|---|---|
| northing (m) | 32926.47 | 33888.66 | 33990.61 | 34804.57 |
| easting (m) | 4874.29 | 3798.01 | 5695.30 | 4807.90 |
| elevation (m) | 485.94 | 486.34 | 437.06 | 452.81 |
| azimuth range (°) | 38.54 - 97.36 | 357.39 - 54.45 | 246.88 - 183.48 | 279.43 - 221.17 |
| mean elevation (°) | 1.83 | 1.79 | 4.71 | 3.80 |
| range gates | 195 (from 700 m every 10 m up to 2640 m) | | | |
| accumulation time (ms) | 500 | | | |
| pulse length (ns) | 200 | | | |

### 2.2 Mast measurements

For this study, we use measurements from four masts. One 100 m mast was located on the NE ridge and a 100 m and two 60 m masts that were located on the SW ridge. All masts are equipped with 3-D ultrasonic anemometers (Gill WindMaster Pro) and temperature sensors (NCAR SHT75) at the heights of 10, 20, 30, 40 and 60 m and 2, 10, 20, 40 and 60 m, respectively.
The 100 m masts also have instruments at 80 and 100 m. Data were acquired at 20 samples per second with a 1 $\mu$s resolution GPS-based time stamp on every sample.





## 3  Flow modeling overview

In this study, long-term simulations of Wagner et al. (2019a, b) are compared to lidar ridge scans to determine the quality of a numerical model over complex terrain. Model simulations were performed with the Weather Research and Forecasting (WRF) model (Skamarock et al., 2008) on three nested domains D1 to D3 with horizontal resolutions of 5 km, 1 km, and 200 m, respectively. The innermost domain D3 is run in large-eddy simulation (LES) mode. The complete model setup including the physical parameterizations that were used is described in detail in Wagner et al. (2019a) and in Wagner et al. (2019b). Two simulations were performed for the whole IOP of the Perdigão 2017 campaign and are run with (WRF_F) and without (WRF_NF) a forest parameterization in the LES domain D3. Without forest parameterization, surface drag is defined by an aerodynamic roughness length $z_0$, which is obtained from the CORINE 2012 land-use data set and converted to land-use types according to Pineda et al. (2004). In the WRF_F run, an additional forest drag term following Shaw and Schumann (1992) is implemented, which decelerates the flow on the lowermost model levels. The forest cover and leaf area index (LAI) are retrieved from the CORINE data set. As no information about the tree height was available for the modeling domains, a randomly uniformly distributed forest height of 30 m $\pm$ 5 m was used. The high resolution aerial scans are only available for a smaller area centered around the measurement site (Figure 1b). A detailed description of the forest parameterization and the differences between the WRF_F and WRF_NF simulations is given in Wagner et al. (2019b).

## 4  Data overview

In the following, data processing methods for the mast, lidar and WRF-LES datasets are described. The measurement datasets are publicly available. The lidar data can be obtained from Menke et al. (2018a) and mast data is available via the NCAR archive in 5 minute and 20 Hz resolution (UCAR/NCAR - Earth Observing Laboratory, 2019a, b).

### 4.1  Mast data

The anemometer data were rotated into a vertical coordinate system and oriented to true North from angles determined by laser multistation scans of each instrument. No issues were determined in the quality control process, so the reported data from the anemometers have been used unedited.

The fans used to aspirate the temperature/relative humidity sensors on the masts occasionally failed during the project. Data from these periods were removed. Also, for some of these sensors, laboratory post-experiment calibrations indicated larger than expected differences from the pre-calibrations (usually less than 0.5 degC and 4%RH). For these sensors, the post-calibrations were applied.





## 4.2 Lidar data

We process the lidar data in three consecutive steps. First, the data are filtered using the method described in section 4.2.1. Next, the measurements of the filtered scans along the ridge trajectories are combined to horizontal winds, see section 4.2.2. Finally, the combined measurements are averaged over 10 minute periods.

### 4.2.1 Filtering

Most commonly, lidar data are filtered by thresholding using the carrier-to-noise ratio (CNR) as a quality indicator. These methods are described by Beck and Kühn (2017) who give a general overview of lidar data filtering approaches and also present highly innovative methods. Here we are proposing a new approach which is based on the assumption that the wind field has a certain degree of continuity. We filter the lidar data in a three-stage process that is applied to each scan: In stage one, the data are filtered based on a moving median value of the LOS velocities measured along each LOS. The median is calculated for a window that stretches over 15 range gates corresponding to a distance of 150 m. All range gates that deviate by a threshold of $3\,\mathrm{m\,s^{-1}}$ from the median are excluded.

In stage two, all measurements that exceed the median of radial velocities along an entire LOS by a threshold value of $6\,\mathrm{m\,s^{-1}}$ are filtered out. Both thresholds were determined by visual inspections of plotted data and tuned to the present values. After each stage, missing range gates are linearly interpolated by the value of the two neighboring range gates in case they have valid values. In a final stage, range gates with valid values that are surrounded by three or more invalid range gates out of the two previous and two following range gates are excluded. These range gates are considered as scatter that are unlikely to have a valid measurement or have a meaningful contribution to the analysis. The first two stages are intended to remove local and global artifacts in the measurements. Finally, all filtering stages are repeated across LOSs in the azimuthal direction.

We demonstrate the performance of this method compared to CNR filters with the thresholds of -24 dB and -27 dB (Figure 2). Our approach recovers more data in the far range of the scans thus extends the range of the scans during periods with low CNR and can remove artifacts caused by e.g. hard targets or second return pulses originating from, for example, a cloud base at a higher elevation. The average availability with our dynamic approach is 91.8% compared to 77.7% (92.2%) with a -24 dB (-27 dB) filter. The high availability of the -27 dB filter is misleading in the sense that this method does not remove all artifacts from the scans (compare Figure 2c).

### 4.2.2 Wind vector reconstruction

The horizontal components of the wind vector ($u$ positive east and $v$ positive north) are reconstructed from measurements of the two WindScanners measuring along the same ridge. The measurements at the 92 ridge trajectory points are combined applying equation 1:

$$\begin{bmatrix} u \\ v \end{bmatrix} = \begin{bmatrix} \sin\phi_1\cos\vartheta_1 & \cos\phi_1\cos\vartheta_1 \\ \sin\phi_2\cos\vartheta_2 & \cos\phi_2\cos\vartheta_2 \end{bmatrix}^{-1} \cdot \begin{bmatrix} V_{r1} \\ V_{r2} \end{bmatrix} \tag{1}$$

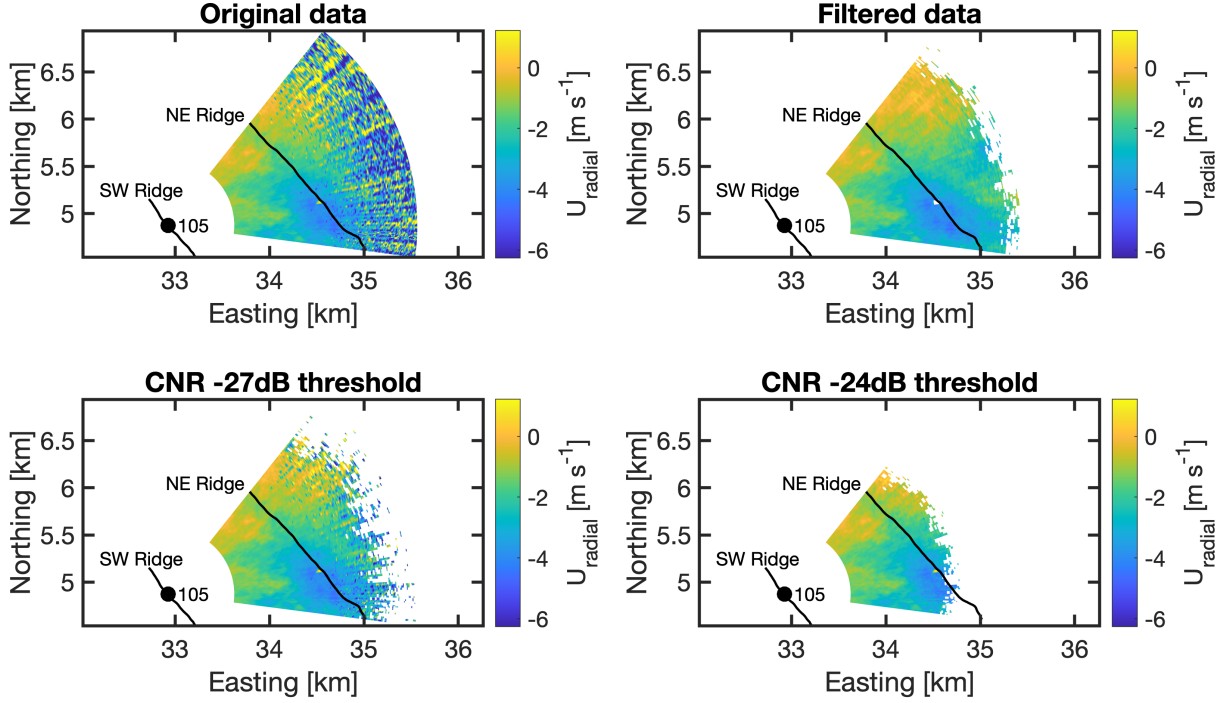

**Figure 2.** Comparison of data recovery with different filters for the 10 minute period starting at May 03, 2017 13:40UTC. a) unfiltered data, b) dynamic filtering approach, c) -24 dB filter, and d) -27 dB filter.

with $V_r$ being the radial or LOS velocities measured by the two WindScanners, $\phi$ the azimuth angles using the geographical convention, i.e. 0° is pointing north and $\phi$ increases clockwise, and $\vartheta$ the elevations angles of the scanners. In this calculation the influence of the vertical wind component $w$ is considered to be negligible since we measured at low elevation angles. We combine 10-minute averaged radial velocity components. Measurement points with less than 10 independent samples are disregarded as well as complete scans with more than 20% invalid data.

### 4.2.3 Data availability

The four WindScanners operated for different periods from March 22 to July 24. Individual system availability in these periods range from 59% to 80% (Table 2). During the IOP, due to the permanent presence of people at the site to aid in the case of a power grid or system failures, the WindScanners' availability is higher (71% to 92%). For dual-Doppler retrievals at the individual ridges, concurrent availability of WS5 and WS6 for the NE ridge and WS7 and WS8 for the SW ridge is required. The combined availability during the IOP is 79% and 51% for the NE and SW ridge, respectively. Simultaneous measurements at both ridges are available for 44% of the period of the IOP. After applying filtering processes as explained in section 4.2.1,



the data availability reduces to 31.6%. When wind speeds are also required to be above $3\,\mathrm{m\,s^{-1}}$ at 80 m height (measured at the mast tse04) 507 10 minute periods are left for the analysis, corresponding to 23% of the IOP period.

**Table 2.** Operation time and data availability of WindScanners. Number in brackets is the number of available 10 minute periods.

| WindScanner | WS5 | WS6 | WS7 | WS8 |
|---|---|---|---|---|
| start of operation | March 27, 16:50 | March 27, 16:50 | March 22, 17:50 | March 27, 16:50 |
| end of operation | June 17, 15:20 | June 17, 09:50 | July 10, 16:50 | July 24, 15:50 |
| scanner availability | 72.8% (2863) | 79.8% (3130) | 58.6% (3094) | 63.2% (3608) |
| scanner availability IOP | 82.2% (1815) | 91.6% (2023) | 70.7% (1562) | 77.0% (1701) |
| combined availability IOP (per ridge) | NE ridge 79.3% (1751) | | SW ridge 51.3% (1133) | |
| combined availability IOP (both ridges) | 44.2% (976) | | | |
| combined availability IOP (after filtering) | 31.6% (698) | | | |
| combined availability IOP (after filtering, $U > 3\,\mathrm{m\,s^{-1}}$) | 23.0% (507) | | | |

## 4.3 Flow model data

Model data of the LES domain D3 is available with a 10 minute output interval. This means that every 10 minutes a snapshot of the simulated meteorological condition is written to the output file. The three-dimensional fields are interpolated linearly in both the horizontal and vertical direction to the lidar ridge scan coordinates. This results in time series of meteorological variables at each lidar scanning point, which can be compared to lidar data.

## 5 Data analysis

### 5.1 Comparison of mast and lidar measurements

The correlation of radial velocities measured by the individual WindScanners and of the reconstructed wind vectors with the sonic wind speeds is calculated. We project the 80 m sonic wind speeds to the lidar LOSs using equation 2.

$$V_{r\_sonic} = u\sin\phi\cos\vartheta + v\cos\phi\cos\vartheta + w\cos\vartheta \tag{2}$$

The sonic data are averaged exactly during the accumulation period (500 ms) at the two closest range gates to the masts that are not affected by the measurement mast structures. These range gates are about 40 m to NW (northwest) and SE (southeast) of the masts. For all WindScanners the correlation coefficient for the LOS measurements are better than 0.994, offsets are less than $0.45\,\mathrm{m\,s^{-1}}$ and slopes deviate by less than 0.04 from 1 (Figure 3). Considering that the measurements are not collocated





and that the measurement volumes of lidars and sonics differ by about two orders of magnitude these correlations can be considered as good.

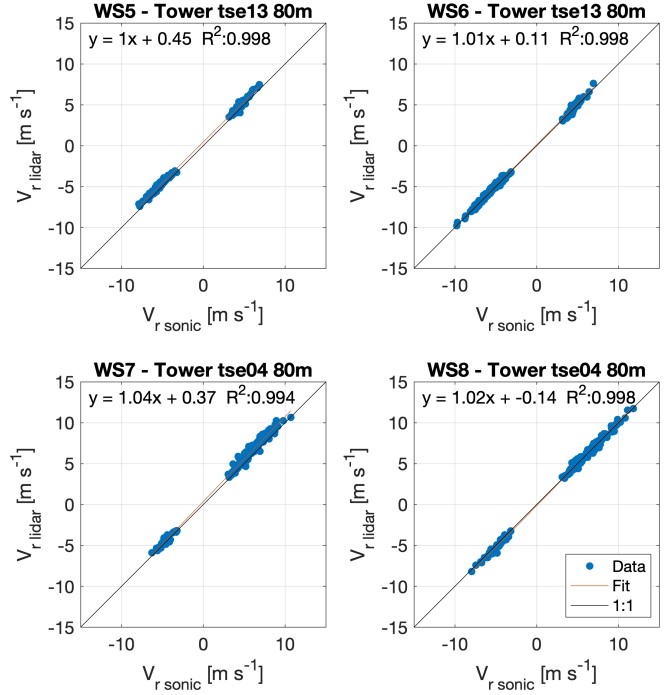

**Figure 3.** Correlation of radial lidar wind speeds with the sonic wind speeds projected to the lidar LOSs. Only southwesterly and northeasterly wind directions are selected for sectors of $\pm 15°$ centered around the transect orientated $54°$ towards north.

The correlation based on the reconstructed wind vectors is calculated for 10-minute averages at all four mast. For the sonic measurements we consider the horizontal wind speed ($U_{\mathrm{hor}} = \sqrt{u^2 + v^2}$) and the wind speed projected to the plane spanned by

5    the two lidars ($U_{\mathrm{proj}} = \sqrt{u_{\mathrm{proj}}^2 + v_{\mathrm{proj}}^2}$). Where the projected wind vector is calculated as:

$$\boldsymbol{U}_{\mathrm{proj}} = \boldsymbol{n} \times (\boldsymbol{U} \times \boldsymbol{n}) \tag{3}$$

with $\boldsymbol{n}$ being the unit normal vector of the plane spanned by the two lidar beams.

The correlation coefficients with the two 80 m sonics are both better than 0.94, with offsets smaller than $0.25\,\mathrm{m\,s^{-1}}$ and slopes close to 1 (1.04 at tower tse04 and 0.94 at tower tse13, Figure 4). At the 60 m masts the correlation of lidar and sonic

10    measurements is lower due to the spatial difference in height. The correlation coefficients at both masts are 0.9. Differences in the correlations of using the projected or the horizontal sonic winds speeds are negligibly small.





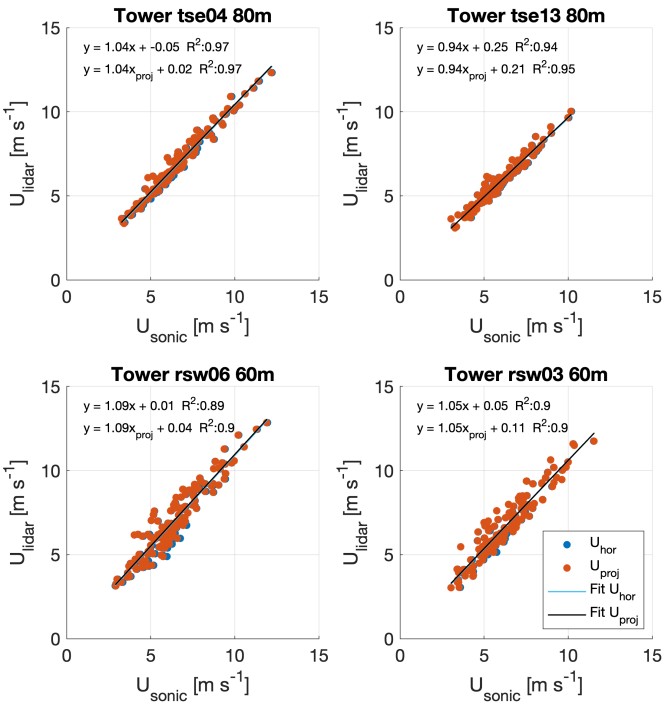

**Figure 4.** Correlation of reconstructed lidar wind speeds with the horizontal sonic wind speeds and the sonic wind speeds projected to the plane spanned by the lidars. Only southwesterly and northeasterly wind directions are selected for sectors of $\pm\ 15°$ centered around the transect orientated 54° towards north.

## 5.2 Observed flow patterns

Considering all available ridge scan periods (507) we find that the mean wind speed is 10% higher at the SW ridge. Relative changes in wind speed along the SW ridge are below 2%. At the NE ridge, the lowest relative wind speeds are found at the terrain dip at 400 m and a change of 7% in mean wind speed is found along the ridge. This picture changes significantly dur-
5    ing specific atmospheric conditions which are analyzed in the following subsections. We segregate the data by the prevailing flow directions from the northeast and the southwest for sectors of $\pm\ 15°$ centered perpendicular to the ridge, orientated at 54° (geographical convention). Furthermore, the data are segregated by the atmospheric stability characterized by the Richardson number ($Ri$) calculated at the upstream mast as defined in Menke et al. (2019b) based on the potential temperature gradient from 20 m to 100 m and the 100 m wind speed. It is not obvious how to define limits for different stability regimes thus we
10    define stable conditions as periods with $Ri > 0$ and unstable conditions as $Ri < 0$. Neutral conditions are only expected to occur during short transition periods.





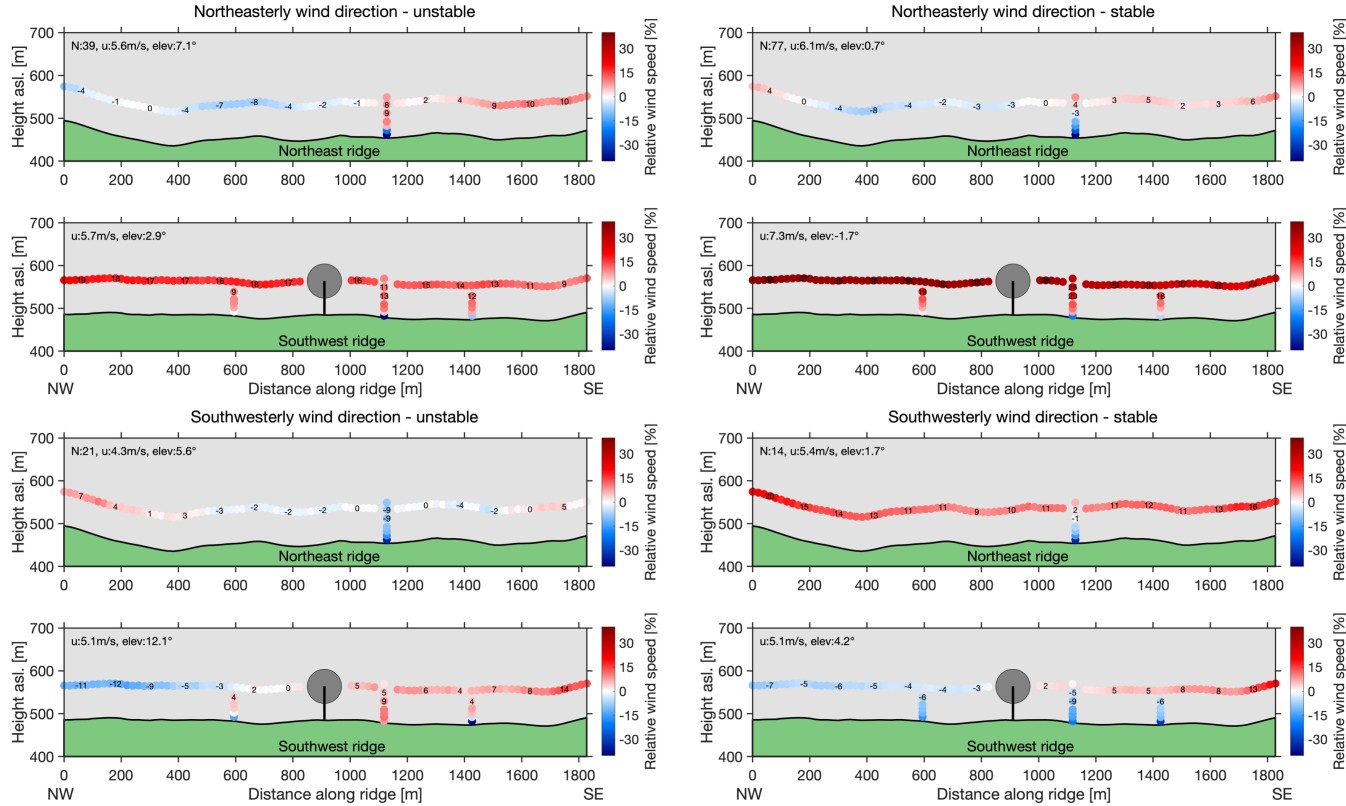

**Figure 5.** Normalized wind speeds measured by the lidars and sonics during different atmospheric conditions. The wind speeds are normalized by the mean along the upstream ridge e.g. for southwesterly wind directions all measurements are normalized with the mean wind speed along the SW ridge.

### 5.2.1 Dependence on wind direction

For southwesterly flows, an increase of more than 20% in relative wind speeds is observed along the SW ridge with higher wind speeds in the southeast (SE) and lower wind speeds in the northwest (NW) (Figure 5). At the NE ridge, for southwesterly flow, increased relative wind speeds of up to 13% are observed at the NW end of the ridge where the elevation is increasing.

5  All values are relative to the mean wind speed along the upstream ridge.

For northeasterly flow, significantly higher wind speeds of about 25% are observed at the SW ridge. Additionally, a change in wind speed along the SW ridge is observed with higher speeds in the NW and lower in the SE which is opposite to the observation under southwesterly flow. For some conditions, the change in relative wind speed is higher than 20%.

We considered these observations as statistically significant as the mean of standard deviations calculated at each point along

10  the ridge is much lower than the observed changes (Table 3).





**Table 3.** Observation from tower tse04 (SW ridge) and tse13 (NE ridge). Turbulence intensity is defined as $TI = \sigma_U \overline{U}_{\text{sonic}}^{-1}$ where $\overline{U}_{\text{sonic}}$ is the mean wind speed and $\sigma_U$ the standard deviation of $U_{\text{sonic}}$. Turbulent kinetic energy is calculated as $e = \frac{1}{2}\left[\overline{u'^2} + \overline{v'^2} + \overline{w'^2}\right]$, where $u'$, $v'$ and $w'$ are the fluctuating parts of the wind vector components as measured by the sonic anemometers. Wind shear and veer are calculated over 60 m (40 m – 100 m). The flow inclination angle $\tau$ is calculated as $\arctan(w\sqrt{u^2 + v^2}^{-1})$, where $u$ and $v$ are the mean horizontal wind vector components, and $w$ the vertical. All averages are taken over 10 minutes. $\overline{U}_{\text{lidar}}$ is the mean of the wind speeds measured by the lidars averaged over the entire ridge. $\overline{\sigma}_{\text{lidar}}$ is the standard deviations also averaged along the ridge. $N$ is the number of available 10-minute periods.

| southwesterly flow | | $\overline{U}_{\text{lidar}}$ $(\text{m s}^{-1})$ | $\overline{\sigma}_{\text{lidar}}/\overline{U}_{\text{lidar}}$ (%) | $\overline{U}_{\text{sonic}}$ $(\text{m s}^{-1})$ | $TI$ (%) | $e$ $(\text{m}^2\text{s}^{-2})$ | shear $(\text{m s}^{-1})$ | veer (°) | $\tau$ (°) | $N$ |
|---|---|---|---|---|---|---|---|---|---|---|
| stable | SW ridge | 5.43 | 3.9 | 5.13 | 11.27 | 0.29 | 0.012 | -0.074 | 4.18 | 14 |
| | NE ridge | 5.75 | 7.6 | 5.41 | 17.33 | 0.63 | 0.011 | -1.036 | 1.75 | |
| unstable | SW ridge | 5.01 | 10.5 | 5.11 | 32.96 | 1.29 | -0.006 | 0.029 | 12.12 | 21 |
| | NE ridge | 4.56 | 13.3 | 4.35 | 43.58 | 1.51 | 0.001 | -0.003 | 5.57 | |
| **northeasterly flow** | | | | | | | | | | |
| stable | SW ridge | 7.66 | 4.9 | 7.33 | 11.96 | 0.52 | 0.016 | -0.073 | -1.74 | 77 |
| | NE ridge | 5.90 | 5.3 | 6.09 | 8.52 | 0.18 | 0.020 | -0.147 | 0.67 | |
| unstable | SW ridge | 5.67 | 10.0 | 5.70 | 30.94 | 1.37 | -0.001 | 0.084 | 2.90 | 39 |
| | NE ridge | 5.10 | 8.5 | 5.60 | 29.15 | 1.24 | -0.001 | -0.040 | 7.09 | |

### 5.2.2 Dependence on atmospheric stability

It is most notable that wind speeds at the downwind ridge are always higher than at the upstream ridge during stable conditions (Figure 5). The mean wind speeds along the downwind ridge measured by the lidars are $1.8\,\text{m s}^{-1}$ higher during northeasterly flow and $0.3\,\text{m s}^{-1}$ for southwesterly flows. This increase in wind speed can most likely be explained by the speed up that is

5  caused by the formation of atmospheric waves during stable conditions (Palma et al., 2019). Moreover, the mast measurements show consistently negative wind shear during stable conditions at both masts and as expected, lower levels of turbulence intensity and energy dissipation (Table 3).

During unstable atmospheric conditions, wind speeds are higher at the SW ridge for both flow directions. Remarkable is also the large flow inclination angles measured by the sonics at the upstream ridges of 12.12° (7.09°) for SW flow (NE flow). The

10  much higher flow inclination angle for SW flow over the SW ridge supports the findings of Menke et al. (2018b) that the wind turbine wake is lifted higher during the day (unstable) than during the night (stable).





**Table 4.** Mean difference of WRF simulations and ridge scans calculated as $(U_{\text{WRF}} - U_{\text{lidar}})\,U_{\text{lidar}}^{-1} \cdot 100$ and averaged along the entire ridge. Correlation coefficient (COR) and root mean square error (RMSE) values for the comparison of WRF data with the ridge scan measurements. The first number states the value for the WRF_NF simulation without forest and in brackets the value for the WRF_F run with forest parameterization. Bold values indicating the best model per parameter. The percentage of lidar observations, which describe the respective flow condition is indicated in the last column.

| **all directions** | | Mean difference (%) | COR | RMSE (m s$^{-1}$) | bias (m s$^{-1}$) | Fraction of used data (%) |
|---|---|---|---|---|---|---|
| all | SW ridge | **6.5** ( -35.2 ) | 0.43 ( **0.49** ) | **2.76** ( 3.43 ) | **-0.07** ( -2.52 ) | 100 |
| | NE ridge | **4.1** ( -32.2 ) | **0.46** ( 0.44 ) | **2.78** ( 2.94 ) | **-0.16** ( -2.09 ) | |
| stable | SW ridge | **-3.6** ( -39.0 ) | 0.40 ( **0.52** ) | **2.68** ( 3.68 ) | **-0.77** ( -2.91 ) | 57.4 |
| | NE ridge | **-9.9** ( -36.1 ) | **0.48** ( 0.48 ) | **2.56** ( 2.90 ) | **-0.83** ( -2.27 ) | |
| unstable | SW ridge | **25.1** ( -29.1 ) | **0.50** ( 0.47 ) | **2.83** ( 2.90 ) | **1.19** ( -1.89 ) | 37.1 |
| | NE ridge | 29.9 ( **-24.2** ) | **0.44** ( 0.42 ) | 3.06 ( **2.86** ) | **1.11** ( -1.69 ) | |
| **southwesterly flow** | | | | | | |
| all | SW ridge | **9.0** ( -36.6 ) | 0.38 ( **0.45** ) | **2.61** ( 2.65 ) | **-0.01** ( -1.96 ) | 9.3 |
| | NE ridge | **10.3** ( -43.5 ) | 0.40 ( **0.43** ) | 3.49 ( **3.06** ) | **0.06** ( -2.41 ) | |
| stable | SW ridge | **0.0** ( -42.4 ) | 0.19 ( **0.46** ) | **2.19** ( 2.99 ) | **-0.51** ( -2.41 ) | 2.8 |
| | NE ridge | **-22.1** ( -56.4 ) | 0.35 ( **0.40** ) | **3.24** ( 4.08 ) | **-1.75** ( -3.55 ) | |
| unstable | SW ridge | **30.4** ( -32.7 ) | 0.43 ( **0.54** ) | 2.84 ( **2.17** ) | **1.08** ( -1.50 ) | 4.1 |
| | NE ridge | 53.8 ( **-39.7** ) | 0.46 ( **0.49** ) | 4.05 ( **2.54** ) | 2.39 ( **-1.85** ) | |
| **northeasterly flow** | | | | | | |
| all | SW ridge | **-6.6** ( -30.8 ) | 0.40 ( **0.41** ) | **2.97** ( 3.87 ) | **-0.96** ( -2.63 ) | 22.9 |
| | NE ridge | **-1.3** ( -23.7 ) | **0.43** ( 0.43 ) | 2.82 ( **2.68** ) | **-0.44** ( -1.61 ) | |
| stable | SW ridge | **-20.1** ( -43.2 ) | 0.43 ( **0.45** ) | **3.20** ( 4.44 ) | **-1.90** ( -3.63 ) | 15.2 |
| | NE ridge | **-19.1** ( -35.3 ) | **0.45** ( 0.45 ) | **2.91** ( 3.09 ) | **-1.39** ( -2.28 ) | |
| unstable | SW ridge | 19.9 ( **-6.2** ) | **0.51** ( 0.48 ) | 2.35 ( **2.22** ) | 0.93 ( **-0.62** ) | 7.7 |
| | NE ridge | 33.7 ( **-0.7** ) | 0.48 ( **0.49** ) | 2.68 ( **1.73** ) | 1.44 ( **-0.25** ) | |

**Table 5.** As in Table 4, but for comparison of WRF simulations with tower T20 (tse04) and T29 (tse13) on the SW and NE ridge, respectively.

| **all directions** | | Mean difference (%) | COR | RMSE (m s$^{-1}$) | bias (m s$^{-1}$) |
|---|---|---|---|---|---|
| all | T20 | 31.0 ( **-23.1** ) | 0.44 ( **0.65** ) | 3.18 ( **2.35** ) | 1.49 ( **-1.11** ) |
| | T29 | **23.1** ( -29.2 ) | 0.46 ( **0.56** ) | 3.03 ( **2.45** ) | **1.07** ( -1.35 ) |



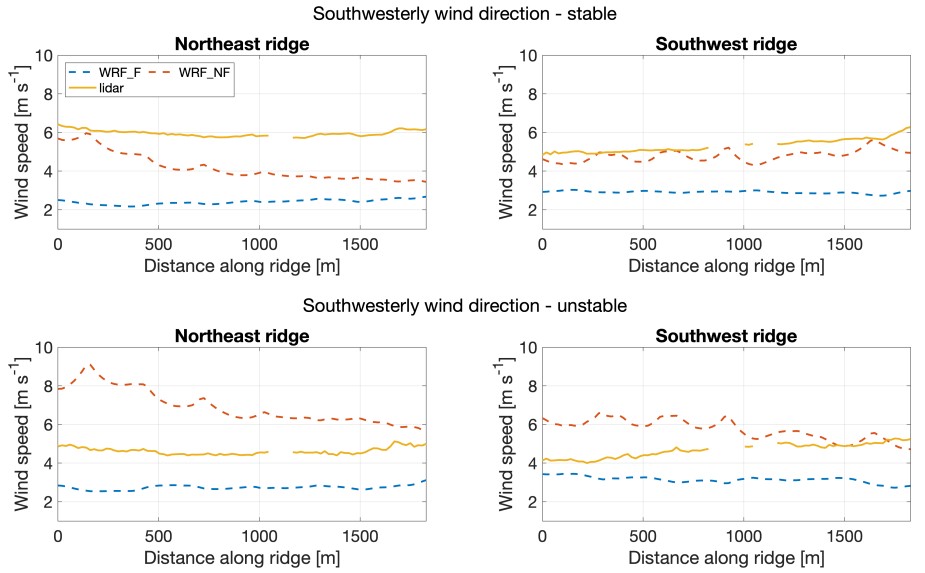

**Figure 6.** Comparison of WRF wind speeds and lidar ridge scan measurements for southwesterly flow segregated into stable and unstable conditions.

### 5.3 Comparison of lidar measurements and simulations

As described in section 3, we compare the ridge scan measurements to the WRF-LES simulations with and without forest drag implementation. Data from both simulations are extracted at the coordinates of the ridge scan points and interpolated to the exact measurement periods in time.

The best agreement, considering all available ridge scan periods, is reached for the WRF_NF simulation without forest drag in terms of mean difference, root mean square error and bias. In this case, the WRF model is overestimating the wind speeds by $6.5\%$ and $4.1\%$ at the SW ridge and NE ridge, respectively (see Table 4). The WRF_F simulation with forest drag implementation underestimates the wind speeds along the ridges by $-35.2\%$ $(-32.2\%)$ at the SW (NE) ridge. This underestimation of simulated wind speeds on the ridge tops was also observed by Wagner et al. (2019b) and is most likely

caused by an over-representation of the forest drag due to incorrect forest coverage on the ridge tops and too high trees in the model. As described in section 3 an average canopy height of 30 m was used, whereas real tree heights obtained from an aerial laser scan in 2015 were in the order of 15 m (see Fig. 1b). The distribution of forested areas in Fig. 1b further indicates that the ridge tops were mostly free of trees, whereas both ridge tops are forested in the model according to the CORINE landuse data set (see Fig. 3 in Wagner et al., 2019b).

Even though the simulation with forest drag underestimates wind speeds at the ridges, it shows improved correlation with the measurements (see Table 4). Correlation coefficients are consistently better for southwesterly wind directions and better or similar to the correlations of the simulation without forest drag for northeasterly flow. A comparison of the same simulations



with multiple meteorological masts across the double ridge along transect southeast (TSE; equal to transect 2 in Fernando et al., 2019) in Wagner et al. (2019b) shows a clear improvement of simulated wind speeds in the WRF_F simulation with forest parameterization. This means that the forest parameterization enhances the simulated flow especially along the slopes of the ridges, where wind speeds are overestimated in the WRF_NF simulation. When comparing the simulations only to the

two 100 m towers tse04 (T20) and tse13 (T29) on the SW and NE ridge, respectively, the WRF_F run underestimates wind speeds at 80 m AGL, but shows improved correlation values and root-mean-square errors (RMSE) (see Table 5 in this paper and Table 4 in Wagner et al., 2019b). The better results for the WRF_F run for the comparison with tse04 and tse13 may be induced by the larger number of samples that are available in the tower data set (about 13500 data points) compared to lidar data (507 points in time) representing a larger spectrum of different meteorological conditions.

Segregating the data into different atmospheric conditions shows that the WRF_NF run performs best under stable atmospheric conditions (Table 4). For unstable conditions, the WRF_F simulation performs better at the NE ridge and for northeasterly wind also at the SW ridge. Considering that the flow is more turbulent under unstable conditions, it can be assumed that more mixing and interaction with the forest is taking place compared to stable conditions during which the forest rather acts as a displacement. For northeasterly winds, the high forest density for the fetch upstream of the NE ridge (see Figure 3 in Wagner

et al., 2019b) might lead to better results of WRF with forest drag.

Figure 6 shows the spatial distribution of averaged wind speeds along the SW and NE ridges during southwesterly flow. The general underestimation of wind speeds in the WRF_F simulation is visible. Disregarding this negative offset, the WRF_F simulation shows spatial changes of wind speed along the ridges that are more similar to changes measured with the lidars compared to stronger gradients along the ridges in the WRF_NF simulation.

Summarizing, we find a high sensitivity of the WRF-LES simulations to the parameterization of surface friction. Adding a forest drag term significantly changes the results. The comparison of the simulations with the lidar ridge scans reveals that the forest drag is too strong on the ridge tops, which results in underestimated wind speeds. Without forest drag, wind speeds are overestimated on average. The comparison of the same simulations with multiple meteorological towers across the double ridge in Wagner et al. (2019b) shows an improvement of the simulated flow with forest parameterization. Also, relative changes

in wind speed along the ridges are more similar for the simulation with forest drag when comparing to the relative changes observed with the WindScanners. This shows that the forest parameterization has a positive effect on simulated wind speeds over Perdigão, but makes clear that the horizontal distribution of forested areas and the tree heights have to be more realistic in future model setups. This will only be possible by using the high resolution aerial lidar scans used for the canopy height estimation in figure 1b, or by introducing better landuse data sets, which include seasonal variability of the canopy layer, e.g.,

caused by forestry and agriculture.

## 6 Conclusions

The present lidar measurements demonstrate the ability of scanning lidars to perform wind resource measurements over large areas in complex terrain. Horizontal mean velocity profiles of 1.8 km length along ridges were retrieved and flow patterns





during specific atmospheric conditions could be identified. The data were collected during the intensive measurement period in May and June 2017 of the Perdigão measurement campaign. An optimized filter for the lidar data is presented which yields in average to 20% more data compared to traditional filtering methods. Correlations of lidar and mast data show good agreement with correlation coefficients of 0.994 or better for line-of-sight velocities and 0.94 or better for horizontal wind speeds. We

found that the lidar elevation angles have a negligible effect on the retrieved horizontal winds.

Considering all lidar measurements we find 10% higher wind speeds at the SW ridge. Segregating the data by wind direction reveals a gradient in wind speeds along the SW ridge, with increasing wind speeds from the NW to SE during southwesterly flows. The effect is reversed for northeasterly flows and amounts for both directions to a change of 14% in relative wind speed along the ridge. During stable atmospheric conditions, wind speeds are found to be highest at the downstream ridge

independent of the wind direction. The mean wind speeds along the downwind ridge measured by the lidars are $1.8 \, \mathrm{m \, s^{-1}}$ and $0.3 \, \mathrm{m \, s^{-1}}$ higher during northeasterly and southwesterly flows, respectively.

The lidar observations have been compared to WRF-LES simulations with and without forest drag implementation. The results show the best agreement, considering all available periods, for the WRF-LES run without forest drag. In that case, the simulation is overestimating the mean wind speeds along the SW ridge by 6.5% and by 4.1% along the NE ridge. Under

unstable atmospheric conditions and northeasterly flow direction, the simulation with forest performs best. The simulation with forest drag has a better correlation coefficient but consistently underestimates the wind speeds at the ridges by 30-40%.

Overall, we conclude that scanning lidar measurements are a valuable tool to asses wind resources in complex terrain. In the future, the system availability has to be improved which was only at 44% for the period investigated in this study. Main factors influencing the availability were software issues, hardware failures and power outages. The comparison of measurements and

flow simulations revealed a high sensitivity of the model to the parameterization of surface friction. In contrast to this study, Wagner et al. (2019b) could show that the forest parameterization considerably improves the boundary layer flow over Perdigão when comparing simulations to meteorological towers across the double ridge. It is assumed that the wrong forest distribution in the model on the ridge tops and the overrated tree heights are the main reason for the poor agreement of WRF_F wind speeds with lidar ridge scans. This shows that aerial lidar scans as used by Boudreault et al. (2015) or more realistic landuse data sets,

including seasonal variability of the canopy distribution, are required as input for flow calculations in the future.



*Data availability.* The WindScanner data for the entire measurement campaign is made available by Menke et al. (2018a). The measurement mast data is made available by NCAR for the 5 minute averaged data (UCAR/NCAR - Earth Observing Laboratory, 2019b) and for high resolution data (UCAR/NCAR - Earth Observing Laboratory, 2019a).

*Author contributions.* Conceptualization, R.M., J.M. and N.V.; Methodology, R.M., J.M., N.V., S.O. and J.W.; Software, R.M. and J.W.;
Validation, R.M., J.M., N.V. and J.W.; Formal Analysis, R.M.; Investigation, R.M., J.M., N.V., S.O. and J.W.; Resources, R.M., J.M., N.V., S.O. and J.W.; Data Curation, R.M., S.O. and J.W.; Writing - Original Draft, R.M., J.M., N.V., S.O. and J.W.; Writing - Review & Editing, R.M., J.M., N.V., S.O. and J.W.; Visualization, R.M.; Project Administration, J.M. and N.V.; Funding Acquisition, J.M. and S.O.

*Competing interests.* The authors declare that they have no conflict of interest.

*Acknowledgements.* We acknowledge the work of everyone involved in the planning and execution of the campaign, in specific we would like
to thank Stephan Voß, Julian Hieronimus (ForWind, University of Oldenburg), Per Hansen and Preben Aagaard (DTU Wind Energy) for their help with the installation of the WindScanners. We are also grateful for the contribution of three WindScanners to the campaign by ForWind. Moreover, only the intensive negotiations of José Carlos Matos, INEGI with local landowners about specific locations for our WindScanners made this research possible. We are grateful to the municipality of Vila Velha de Ródão, landowners who authorized installation of scientific equipment in their properties, the residents of Vale do Cobrão, Foz do Cobrão, Alvaiade, Chão das Servas and local businesses who kindly
contributed to the success of the campaign. The space for operational centre was generously provided by Centro Sócio- Cultural e Recreativo de Alvaiade in Vila Velha de Rodão. We thank the Danish Energy Agency for funding through the New European Wind Atlas project (EUDP 14-II). R. Menke has been supported during the execution of this work by NCAR's Advanced Study Program fellowship.



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
