# Peer review of "Multi-lidar wind resource mapping in complex terrain"

_Wind Energy Science, 2019_

## Referee Comment (RC1) · Anonymous Referee #1 · 26 Dec 2019

**Multi-lidar wind resource mapping in complex terrain**

*Robert Menke, Nikola Vasiljević ́, Johannes Wagner, Steven P. Oncley, and Jakob Mann*

**REVIEW**

**GENERAL COMMENT:**

The manuscript analyzes data from the Perdigão 2017 campaign to show measurements from sonic anemometers, lidars, and compare those with modeled data.

The use of novel observational techniques, especially in complex terrain, is of great interest for the wind energy community. The technical description of the observations used is well-detailed (maybe even a bit too much), and the plots and figures are generally well-made.

However, the purpose of including modeling data (with a rather poor match with the observations) in this study is not made sufficiently clear in the manuscript, especially when considering the title, abstract, and introduction. As a consequence, the reader can be a bit lost in terms of the main goal and novelty of this piece of literature.

Along these lines, in order to create a more coherent story, the authors shall spend some time adding additional sentences throughout the manuscript describing the meaning and/or causes for the results shown, rather than purely describing the data plotted in the figures or listed in the tables.

**SPECIFIC COMMENTS:**

1. P.1 l.6: "we found that for different flow conditions on average […]" is not clear for a reader that has not read the whole paper yet.
2. P.1 l.7: "depending on the atmospheric conditions" is also too vague.
3. I think the story you are trying to tell in the abstract is missing some pieces. According to the first part of the abstract, your goal in this study is to demonstrate that scanning lidars can be used to measure wind in complex terrain. And you re-state this at the very end. However, you do not mention the comparison of the lidar measurements with other instruments to validate your thesis. Then, you mention you simulate the wind flow with WRF-LES to check whether it can represent well the wind flow by comparison with the lidars. And this is not mentioned at all in the title. Please clarify what the actual and final goal (and novelty) of this work is.
4. P.1 l.17: "cheaper" seems too much of a strong opinion to me. It actually depends.
5. P.2 l.5 change to "to assess" or "in assessing".
6. P.2 l.10 "use" instead of "present".
7. P.2 l.13: which parallel ridges? Do not assume the reader is familiar with the campaign: you haven't described it yet.

8. Introduction: the goal of the study is still not sharp clear. From how this reads, you are plotting data measured by the lidars, and comparing with modeled data. What is the advancement provided by this study? How does this relate with the title of the paper?

9. P.2 l.20-24: use either all "Section" or "section" throughout.

10. Figure 1: can you please center panel c in the figure?

11. Figure 1: please only use either "a.s.l." or "asl."

12. Section 2.1.1: most of this paragraph describes how the 2015 design was chosen. Instead, I would prefer the focus to be on the 2017 design. And refer to the 2015 design simply with something like "By extending the design of the 2015 campaign (reference), …".

13. P.4 l.25: again, I don't think we need to know how the 2015 were being generated.

14. P.4 l.26: "For these reasons, the initial deployment of instruments was complicated and time-consuming." is probably not needed in a scientific paper.

15. Section 2.1: I feel like this whole section has a bit too much details that would be beneficial for a technical report, not so much for the main body of a scientific publication. Please consider moving some details to an appendix/SI. Also, after filtering the relevant information, the division in subsections might not be needed anymore.

16. Table 1: the names of the WindScanners in the table do not match those shown in the maps in Figure 1. Please correct and be consistent.

17. P.5 l. 19: add "above ground level" after the height of the instruments on the masts.

18. Section 2.2: you have not mentioned the sonic anemometers anywhere in the abstract or introduction, so now this feels a bit confusing. Please clarify earlier in the text.

19. P.6 l. 18-20: this is a repetition of what later reported in the data availability section, and as such can be safely removed.

20. P.7 l.11: why did you choose 15 range gates and not a different number?

21. Figure 2: to make the panels larger, you can consider including only one color bar for the whole figure, instead of the four shown now, as they are all the same. Same comment with axis labels. Also, include panel names (a, b, c, d) in the figure.

22. Section 4.3: it seems like lidar data are computed as 10-minute average, while the LES data are instantaneous data. If this is the case, please comment on how this difference can impact your comparison between observations and modeled data.

23. Section 4.2.3: after reading the section, it is still not clear to me which timeframe you are using for your analysis. Only the IOP, or the full period of overlap among all lidars? Please clarify.

24. P.10 l. 13: do you mean "lidar data" here?

25. Figure 3: what is the temporal period of the comparison shown? Why were only certain wind directions chosen? This should also be specified in the main text, and not only in the figure caption. Also, please make the panels larger.

26. Figure 4: the titles of the plots are not consistent with those of Figure 3. Please, larger panels. Why are two regression lines shown in each plot? This is not described in the text.

27. Section 5.1: I think it is important, at the end of the section, to be explicit about the overall purpose of the comparison (which I guess was to validate the lidar data?).
28. Figure 5: only one color bar and larger panels, please. Some panels have a "N" label in the top-left corner, some have not: please be consistent.
29. Section 5.2.1 describes the data, but do you have any possible explanation on *why* what you describe is happening?
30. Table 3: over which time interval is TI (i.e. mean and std) calculated? Same question for TKE. Add "AGL" after "100 m". Is veer really in degrees? Or degrees per meter?
31. P. 13 l.7: where is turbulence dissipation shown in the table?
32. P.15: the correlation is quite poor, even for the "best" model setup. This should be pointed out in the text and critically explained.
33. P.15 l.12: so why you didn't pick 15m for the modeled tree height?
34. P. 17 l. 28: rephrase as "In the future, the system availability, which was only at 44% for the period investigated in this study, has to be improved."
35. Data availability: "high-resolution".
36. References: please make sure that all listed references are in the same format. Some titles have capital letters for each word, some have not. Some publications have the DOI not listed.

---

## Referee Comment (RC2) · Anonymous Referee #2 · 2 Jan 2020

**Multi-lidar wind resource mapping in complex terrain**

by
Robert Menke et al.

**Review**

**General comments**

The manuscript „Multi-lidar wind resource mapping in complex terrain" by Robert Menke et al. presents dual Doppler lidar measurements from the Perdigao 2017 campaign and compares them to combined mesoscale and large eddy simulations on the basis of ten minute averages. The presented lidar scan pattern along a line with constant height above a ridge is novel and interesting and could possibly serve in applications for resource assessment in complex terrain in the future.

Nevertheless, there are major objections with the current status of the manuscript as a scientific paper. The authors need to add a clear scientific objective and rewrite the manuscript following a red line to answer the research question. Therefore the current knowledge gaps need to be clarified in the introduction, too. The structure of the manuscript needs revision to follow the IMRAD scheme. The figures are well made, some corrections are stated in the specific comments.

The role of the presented simulations is unclear, since they are just introduced briefly and the agreement to the measurements is quite bad. The interesting findings of wind speed differences over the ridges should not just be described in the text but better discussed. The authors should consider to focus on the validation of the lidar scans using the available in situ measurements on the met masts or on improving the simulations on the basis of all presented flow measurements and a realistic forest representation.

**Specific comments and technical corrections**

1. Better define terms or choose just one for similar meanings: scanning lidar, WindScanner, Multi-lidar, dual-Doppler, etc.
2. Please proofread the manuscript on use of times.
3. P2L5, examples for current usage of long range lidar: please precise type of lidar applications, i.e. for single wind turbine wakes no large areas have to be scanned. Please limit references to some recent papers. Further state of the art applications of scanning long range lidar worth mentioning here are scanning lidar based wind/power forecasts and research on wind farm wakes.
4. Sec 2.1.1: shorten description of 2015 campaign since not used here. Focus on setup used in this article. Better explain and justify statements made (e.g. laser beams need to be chosen "as low as possible").
5. P4L16: $\cos(5°) \approx 0.996$, please correct.
6. Sec 2.1.2: Focus on relevant information. Type of lidar power supply not relevant, limit information to the fact of disturbances in power supply.
7. Fig 1: Please use consistent naming for met masts, lidars, etc. Here lidars are named 105, 106, etc. In Table 1 names are WS5, WS6, etc.
8. Fig 1: Please add information about the used coordinate system (PT-TM06/ETRS89) in the text.
9. Sec 2.1.3: Please shorten description of the networking schemes etc. to the relevant information (offsets and usage of GPS time). Add information on spatial averaging along the scan trajectory resulting from continuous scanning and in beam direction resulting mainly from the pulse length. Basic information like the type of the used lidar systems is missing.
10. P5L6: Please specify descriptions of setup. "Range gates were placed every 10 m *along the laser beam ...*".
11. P5L15: unit of first numbers missing! 0.42 s pm 1.03 s or (0.42 pm 1.03) s
12. Sec 2.2: Information on humidity sensors mentioned in the manuscript is missing here!

13. P5L20: "The 100 m masts also have instruments at 80 and 100 m." What kind of instruments are those? Is this information relevant here?
14. Sec 3: If you decide to show simulation results please describe the setup in more detail. It is ok to reference to another detailed description but all basic information needed to understand this work should be included.
15. P6L17-19: Move information on data availability to section "Data availability"
16. P6L21: Please describe the vertical coordinate system and relate it to the coordinates used for lidar data and simulation data.
17. Sec 4.2.1: Has the filter method used here been introduced before? Has it been validated? Is the assumption of the "certain degree of continuity" of the atmosphere and are the chosen thresholds appropriate here? In Figure 2 this approach is called "dynamic filtering approach". Please use consistent naming. Why is the approach dynamic when thresholds are static and chosen manually?
18. Fig 2: Labels a), b) etc. are missing. b) is called "filtered data" in the title and "dynamic filtering approach" in the caption. The order of c) and d) is swapped in the caption. Please correct.
19. Sec 4.3: The information from this short section should be moved to Sec. 3.
20. Sec 5.1: This section is said to contain results but starts with methods (c.f. comment on IMRAD-structure above).
21. Fig 4: Please specify lidars used for single subplots.
22. Fig 4: U_hor and U_proj not introduced in the text. Markers and fits could not be distinguished.
23. Sec 5.2: Findings described are not referenced to the relating figure.
24. P11L7: Please move methodology for the retrieval of atmospheric stability to methods section.
25. Fig 5: Please define N, u and elev given in the plots.
26. Fig 6: Different line styles are suggested for both WRF conditions to support readability in black and white printing.
27. P16L28: Calling the aerial laser scanner measurements "lidar" might confuse the reader. Please distinguish between the terms for wind lidar and lidar for distance measurements.
28. P16L25 and following: It could be expected that numerical simulations without forest parametrization and with a non-realistic forest parametrization will lead to poor results. I suggest simulations with a realistic forest parametrization or excluding simulations with a change of the focus of the manuscript (see comment above).
29. Sec 6: The major part of this section is a summary, not a conclusion. Please rewrite.

---

## Author Comment (AC1) · 10 Apr 2020

Please find our responses to the reviewers in the attached document.

Please also note the supplement to this comment:
https://www.wind-energ-sci-discuss.net/wes-2019-85/wes-2019-85-AC1-supplement.pdf
* * *

---

## Author Response (AR1)

**Response to reviewer 1**

Dear Anonymous Reviewer 1,

we highly appreciate your feedback. It helped us to improve the manuscript. Below we comment on your suggestions in detail.

***GENERAL COMMENT:***

*The manuscript analyzes data from the Perdigão 2017 campaign to show measurements from sonic anemometers, lidars, and compare those with modeled data.*
*The use of novel observational techniques, especially in complex terrain, is of great interest for the wind energy community. The technical description of the observations used is well-detailed (maybe even a bit too much), and the plots and figures are generally well-made.*

*However, the purpose of including modeling data (with a rather poor match with the observations) in this study is not made sufficiently clear in the manuscript, especially when considering the title, abstract, and introduction. As a consequence, the reader can be a bit lost in terms of the main goal and novelty of this piece of literature.*

*Along these lines, in order to create a more coherent story, the authors shall spend some time adding additional sentences throughout the manuscript describing the meaning and/or causes for the results shown, rather than purely describing the data plotted in the figures or listed in the tables.*

Thank you for your comments. We revised the entire manuscript taking into account your comments. Most importantly, we added a Discussion section with the focus on the interpretation of the observed flow structures and the limitation of the WRF-LES simulation. Please find detailed information on our changes below your specific comments and in the attached marked-up version of the manuscript.

***SPECIFIC COMMENTS:***

1. *P.1 l.6: "we found that for different flow conditions on average [...]"is not clear for a reader that has not read the whole paper yet.*

   See answer for comment 3.

2. *P.1 l.7: "depending on the atmospheric conditions" is also too vague.*

   See answer for comment 3.

3. *I think the story you are trying to tell in the abstract is missing some pieces. According to the first part of the abstract, your goal in this study is to demonstrate that scanning lidars can be used to measure wind in complex terrain. And you re- state this at the very end. However, you do not mention*

*the comparison of the lidar measurements with other instruments to validate your thesis. Then, you mention you simulate the wind flow with WRF-LES to check whether it can represent well the wind flow by comparison with the lidars. And this is not mentioned at all in the title. Please clarify what the actual and final goal (and novelty) of this work is.*

Thank you for these detailed comments on the abstract. We reformulated the whole abstract.

4. *P.1 l.17: "cheaper" seems too much of a strong opinion to me. It actually depends.*

   „Cheaper" has been replaced by „can be cost-effective".

5. *P.2l.5 change to "to assess" or "in assessing".*

   Corrected.

6. *P.2 l.10 "use" instead of "present".*

   Changed.

7. *P.2 l.13: which parallel ridges? Do not assume the reader is familiar with the campaign: you haven't described it yet.*

   The sentence has been reformulated. It now reads: "…of wind resources along two ridges, which favorable sites for wind turbines sides, at the Perdigão site.".

8. *Introduction: the goal of the study is still not sharp clear. From how this reads, you are plotting data measured by the lidars, and comparing with modeled data. What is the advancement provided by this study? How does this relate with the title of the paper?*

   The introduction has been reviewed and describes now clearer the study goals.

9. *P.2 l.20-24: use either all "Section" or "section" throughout.*

   Corrected.

10. *Figure 1: can you please center panel c in the figure?*

    Done.

11. *Figure 1: please only use either "a.s.l." or "asl."*

    Corrected.

12. *Section 2.1.1: most of this paragraph describes how the 2015 design was chosen. Instead, I would prefer the focus to be on the 2017 design. And refer to the 2015 design simply with something like "By extending the design of the 2015 campaign (reference), ...".*

Thanks for the suggestion. The paragraph has been reformulated accordingly.

13. *P.4 l.25: again, I don't think we need to know how the 2015 were being generated.*

This information has been removed.

14. *P.4 l.26: "For these reasons, the initial deployment of instruments was complicated and time-consuming." is probably not needed in a scientific paper.*

The content has been removed.

15. *Section 2.1: I feel like this whole section has a bit too much details that would be beneficial for a technical report, not so much for the main body of a scientific publication. Please consider moving some details to an appendix/SI. Also, after filtering the relevant information, the division in subsections might not be needed anymore.*

Thank you for pointing this out. We restructured and reformulated the section.

16. *Table 1: the names of the WindScanners in the table do not match those shown in the maps in Figure 1. Please correct and be consistent.*

The naming convention is now consistent throughout the manuscript.

17. *P.5 l.19: add "above ground level" after the height of the instruments on the masts.*

AGL has been added to the heights of mast-mounted instruments.

18. *Section 2.2: you have not mentioned the sonic anemometers anywhere in the abstract or introduction, so now this feels a bit confusing. Please clarify earlier in the text.*

The use of sonic measurements for validation purposes of the scanning lidar measurements is now mentioned in the abstract and introduction.

19. *P.6 l. 18-20: this is a repetition of what later reported in the data availability section, and as such can be safely removed.*

The content has been removed.

20. *P.7 l.11: why did you choose 15 range gates and not a different number?*

The window size of the moving median such as the thresholds used by the filter have been intensely tuned and tested for the present dataset and has been found to work well in different atmospheric conditions. Different approaches (without success) have been tested to remove fixed thresholds or window sizes from the method.

21. *Figure 2: to make the panels larger, you can consider including only one color bar for the whole figure, instead of the four shown now, as they are all the same. Same comment with axis labels. Also, include panel names (a, b, c, d) in the figure.*

Thank you, we changed the figure according to the suggestions.

22. *Section 4.3: it seems like lidar data are computed as 10-minute average, while the LES data are instantaneous data. If this is the case, please comment on how this difference can impact your comparison between observations and modeled data.*

Yes it's correct, that LES data are instantaneous whereas lidar data are 10-minute averages. We think that the usage of 10-minute averaged LES data would improve the comparison with lidar observations. However, in our opinion the forest parameterization including correct tree heights and horizontal distribution of forested areas across the ridges has a larger impact on the comparison of lidar and LES data than different time averaging intervals. This is only our appreciation and should be investigated in further studies.

23. *Section 4.2.3: after reading the section, it is still not clear to me which timeframe you are using for your analysis. Only the IOP, or the full period of overlap among all lidars? Please clarify.*

The text has been modified to clarify that only periods measured during the IOP period are used. The section reads now as follows: "…. For the analysis, we only use measurement of the IOP period, due to the higher data availability, and removed periods with wind speeds below 3 m s$^{-1}$ at 80 m height (measured at the mast tse04) which leaves 507 10 minute periods, corresponding to 23% of the IOP period."

24. *P.10 l. 13: do you mean "lidar data" here?*

As page 10 doesn't have thirteen lines we assume that this comment refers to line 13 on page 9. There the term "sonic data" was used intentionally and correctly. We averaged the sonic data, which is available continuously with a temporal resolution of 20 Hz, for the periods of 500 ms during which the SLs measured closest to the masts. The wording has been improved to make this clearer in the manuscript.

25. *Figure 3: what is the temporal period of the comparison shown? Why were only certain wind directions chosen? This should also be specified in the main text, and not only in the figure caption. Also, please make the panels larger.*

The information has been added to the caption and the main body of the manuscript: "For this comparison only measurements from IOP are selected, and measurements are limited to the prevailing wind directions (±15° centered around the transect orientated 54° towards north) to minimize the effects of mast wind shadow and to be consistent with the data fraction used for the further analysis." The figure has been increased in size.

26. *Figure 4: the titles of the plots are not consistent with those of Figure 3. Please, larger panels. Why are two regression lines shown in each plot? This is not described in the text.*

Figure 3 and figure 4 cannot be compared directly. Figure 3 shows the correlation of radial velocities of the four SLs and the 80 m sonics. Figure 4 shows the correlation of reconstructed wind speeds of the SLs and the horizontal/projected wind speed measured by the sonics at 60 m and 80 m. This is the reason for the different non-consistent titles of the plots. The two regression lines show the correlation of the reconstructed lidar wind speed with the horizontal wind speed and the wind speed projected to the plane spanned by the SLs measured by the sonics anemometers. This has been made clearer in the manuscript.

27. *Section 5.1: I think it is important, at the end of the section, to be explicit about the overall purpose of the comparison (which I guess was to validate the lidar data?).*

Thanks for this suggestion. A clarifying paragraph of the purpose of the comparison has been added to section 5.1.

28. *Figure 5: only one color bar and larger panels, please. Some panels have a "N" label in the top-left corner, some have not: please be consistent.*

We updated the figure and it shows now only one color bar as suggested. The label "N" specifies the number of 10-minute periods used per case. The number is identical per case; thus, it is only shown once per case in the plot of the northwest ridge and not repeated in the plot of the southwest ridge.

29. *Section 5.2.1 describes the data, but do you have any possible explanation on why what you describe is happening?*

We agree that the manuscript missed discussion and explanation of the observed flow structures. We added a whole discussion section (Section 6 of the revised manuscript) to address these points.

30. *Table 3: over which time interval is TI (i.e. mean and std) calculated? Same question for TKE. Add "AGL" after "100 m". Is veer really in degrees? Or degrees per meter?*

Thanks for catching these points. They have been corrected in the manuscript.

31. *P.13 l.7: where is turbulence dissipation shown in the table?*

The term "turbulence dissipation" has been by mistake used for "turbulent kinetic energy". This has been corrected in the manuscript.

32. *P.15: the correlation is quite poor, even for the "best" model setup. This should be pointed out in the text and critically explained.*

It is right that the correlations are quite poor. We explain this, however, on page 15 and 16 by means of Wagner et al. 2019b. The forest drag on the ridges is too strong due to too high tree heights and due to a wrong horizontal forest distribution on the ridge tops, which is induced by insufficient landuse data sets. Further model simulations with better forest distribution and more realistic tree heights need to be done in the future to investigate this effect.

33. *P.15 l.12: so why you didn't pick 15m for the modeled tree height?*

The tree height of 30 m was chosen when the simulation setup was configurated and no information about actual tree height was available.

34. *P. 17 l. 28: rephrase as "In the future, the system availability, which was only at 44% for the period investigated in this study, has to be improved."*

The sentence has been rephrased according to the suggestion.

35. *Data availability: "high-resolution".*

Corrected.

36. *References: please make sure that all listed references are in the same format. Some titles have capital letters for each word, some have not. Some publications have the DOI not listed.*

DOIs have been added to each reference if available. Capitalized titles could not be identified.

**Response to reviewer 2**

Dear Anonymous Reviewer 2,

we highly appreciate your feedback. It helped us to improve the manuscript. Below we comment on your suggestions in detail.

**General comments**

*The manuscript „Multi-lidar wind resource mapping in complex terrain" by Robert Menke et al. presents dual Doppler lidar measurements from the Perdigão 2017 campaign and compares them to combined mesoscale and large eddy simulations on the basis of ten minute averages. The presented lidar scan pattern along a line with constant height above a ridge is novel and interesting and could possibly serve in applications for resource assessment in complex terrain in the future. Nevertheless, there are major objections with the current status of the manuscript as a scientific paper. The authors need to add a clear scientific objective and rewrite the manuscript following a red line to answer the research question. Therefore, the current knowledge gaps need to be clarified in the introduction, too. The structure of the manuscript needs revision to follow the IMRAD scheme. The figures are well made, some corrections are stated in the specific comments.*

*The role of the presented simulations is unclear, since they are just introduced briefly and the agreement to the measurements is quite bad. The interesting findings of wind speed differences over the ridges should not just be described in the text but better discussed. The authors should consider to focus on the validation of the lidar scans using the available in situ measurements on the met masts or on improving the simulations on the basis of all presented flow measurements and a realistic forest representation.*

We revised the entire manuscript with a focus to clarify the study's objectives and having a clearer structure. The revised manuscript follows now the IMRAD scheme. Please see for details our comments to your specific comments below and the attached marked-up version of the manuscript. Most importantly, we added an entire discussing section focusing on the interpretation of the observations and the role and limitations of the WRF-LES simulation.

**Specific comments and technical corrections**

1. *Better define terms or choose just one for similar meanings: scanning lidar, WindScanner, Multi-lidar, dual-Doppler, etc.*

   Thanks for this comment. We use now consistently the term "scanning lidar" throughout the manuscript.

2. *Please proofread the manuscript on use of times.*

   We proofread the entire manuscript and corrected the use of times.

3. *P2L5, examples for current usage of long range lidar: please precise type of lidar applications, i.e. for single wind turbine wakes no large areas have to be scanned.*

*Please limit references to some recent papers. Further state of the art applications of scanning long range lidar worth mentioning here are scanning lidar based wind/power forecasts and research on wind farm wakes.*

This paragraph has been updated it now reads as follows: "Moreover, many studies utilized the scanning capability to measure wind fields over large areas for wind energy purposes in assessing, for example, wind turbine wakes (Trujillo et al., 2011; Iungo et al., 2013; Bodini et al., 2017; Menke et al., 2018b), the inflow towards wind turbines (Mikkelsen et al., 2013; Simley et al., 2016; Mann et al., 2018), the influence of surface and terrain features on the flow (Lange et al., 2016; Mann et al., 2017) and atmospheric phenomena such as gravity waves (Palma et al., 2019)."

4. *Sec 2.1.1: shorten description of 2015 campaign since not used here. Focus on setup used in this article. Better explain and justify statements made (e.g. laser beams need to be chosen "as low as possible").*

   The description of the section now solely focuses on the 2017 campaign.

5. *P4L16: cos(5°) ≈ 0.996, please correct.*

   Corrected.

6. *Sec 2.1.2: Focus on relevant information. Type of lidar power supply not relevant, limit information to the fact of disturbances in power supply.*

   Unnecessary information has been removed from this section.

7. *Fig 1: Please use consistent naming for met masts, lidars, etc. Here lidars are named 105, 106, etc. In Table 1 names are WS5, WS6, etc.*

   Corrected. We are referencing the SLs now by the 105, 106, … convention throughout.

8. *Fig 1: Please add information about the used coordinate system (PT-TM06/ETRS89) in the text.*

   The coordinate system is mentioned in the caption of the figure.

9. *Sec 2.1.3: Please shorten description of the networking schemes etc. to the relevant information (offsets and usage of GPS time). Add information on spatial averaging along the scan trajectory resulting from continuous scanning and in beam direction resulting mainly from the pulse length. Basic information like the type of the used lidar systems is missing.*

   The description of the network scheme has been shortened and information about spatial averaging and other basic information has been added.

10. *P5L6: Please specify descriptions of setup. "Range gates were placed every 10 m along the laser beam ...".*

The description of the scanning scenario has been expanded including a definition of the range gate term.

11. *P5L15: unit of first numbers missing! 0.42 s pm 1.03 s or (0.42 pm 1.03) s*

Corrected.

13. *Sec 2.2: Information on humidity sensors mentioned in the manuscript is missing here!*

The NCAR SHT75 sensor is a combined temperature and humidity sensor. This information was missing in this section and has been added. Thanks for pointing this out.

14. *P5L20: "The 100 m masts also have instruments at 80 and 100 m." What kind of instruments are those? Is this information relevant here?*

The instruments at 80 and 100 m are sonic anemometers and temperature /humidity sensors. The data of the sensors at these heights is used in this manuscript and those important to mention. We clarified the instrument type in the manuscript.

15. *Sec 3: If you decide to show simulation results please describe the setup in more detail. It is ok to reference to another detailed description, but all basic information needed to understand this work should be included.*

We significantly expanded the description of the simulation. Please find the changes in the attached marked up version of the manuscript.

16. *P6L17-19: Move information on data availability to section "Data availability"*

The information has been moved to the "Data availability" section. Thanks for pointing this out.

17. *P6L21: Please describe the vertical coordinate system and relate it to the coordinates used for lidar data and simulation data.*

The coordinate system has been specified. The manuscript reads now as follows: "The anemometer data are rotated into a vertical coordinate system (i.e. w is aligned with the vertical axis of the local coordinate, PT-TM06/ETRS89, system which is also used for the lidar data) and oriented to true North from angles determined by laser multistation scans of each instrument."

18. *Sec 4.2.1: Has the filter method used here been introduced before? Has it been validated? Is the assumption of the "certain degree of continuity" of the atmosphere and are the chosen thresholds appropriate here? In Figure 2 this approach is called*

*"dynamic filtering approach". Please use consistent naming. Why is the approach dynamic when thresholds are static and chosen manually?*

To our knowledge, this filtering method has not been introduced before. However, it is based on a similar, less complex filtering approach that was developed for a different lidar dataset that has been not been published. The thresholds and the underlying assumption of continuity in the wind field have been intensely tuned and tested for the present dataset and has been found to work well in different atmospheric conditions. Different approaches, without success, have been tested to remove fixed thresholds from the method. The use of the term "dynamic" was taken up as the filtering method, due to its running averages windows, adjusts to certain parts of the measured wind field. We agree that the term can be misleading and removed it from the manuscript.

19. *Fig 2: Labels a), b) etc. are missing. b) is called "filtered data" in the title and "dynamic filtering approach" in the caption. The order of c) and d) is swapped in the caption. Please correct.*

   The figure has been corrected.

20. *Sec 4.3: The information from this short section should be moved to Sec. 3.*

   The section has been integrated into section 3 as suggested.

21. *Sec 5.1: This section is said to contain results but starts with methods (c.f. comment on IMRAD-structure above).*

   Thanks for this comment. We moved the description of methods from this section to section 4.1.

22. *Fig 4: Please specify lidars used for single subplots.*

   The lidars are now specified in the caption of the figure.

23. *Fig 4: U_hor and U_proj not introduced in the text. Markers and fits could not be distinguished.*

   U_hor and U_proj are introduced in section 4.1. The markers are indeed difficult to distinguish as the difference between U_hor and U_proj is neglectable what we are highlighting here.

24. *Sec 5.2: Findings described are not referenced to the relating figure.*

   The findings described in section 5.2 are not shown in any figure that is part of the manuscript. We added a note specifying this to the manuscript.

25. *P11L7: Please move methodology for the retrieval of atmospheric stability to methods section.*

Thanks for this suggestion. The content has been moved to section 4.1.

26. *Fig 5: Please define N, u and elev given in the plots.*

   The definitions have been added.

27. *Fig 6: Different line styles are suggested for both WRF conditions to support readability in black and white printing.*

   The line styles have been changed to improve the black and white printing readability.

28. *P16L28: Calling the aerial laser scanner measurements "lidar" might confuse the reader. Please distinguish between the terms for wind lidar and lidar for distance measurements.*

   The term "aerial laser" is now consistently used when referring to lidar distance measurements.

29. *P16L25 and following: It could be expected that numerical simulations without forest parametrization and with a non-realistic forest parametrization will lead to poor results. I suggest simulations with a realistic forest parametrization or excluding simulations with a change of the focus of the manuscript (see comment above).*

   Generally, forest is considered in simulations by increasing the surface friction in forested areas and special forest parameterization is not considered as a standard method. Thus, using a simulation without specific forest drag parametrization can be considered as a reference case. In our view, it is not obvious which result can be expected from either of the two simulations. Even though the forest height in the simulation with parametrization is too high, we could show that the correlation of simulation results and measurements has been improved which can be valuable for future simulation setup. Due to limited resources, we are unfortunately not able to perform further simulations.

30. *Sec 6: The major part of this section is a summary, not a conclusion. Please rewrite.*

   The conclusion section has been entirely rewritten. Please find the updated version in the attached marked up version revised manuscript.

[revised manuscript text omitted]

---

## Referee Report (RR1)

Thank you for addressing most of my comments from the previous review. Please find below additional minor suggestions.

*Note: references to page and lines in this list uses the marked-up version of the paper.*

1. P.2 l.19: avoid repetition of "site".
2. P.2 l.28: I still insist that it is important to emphasize the outcome of the comparison between observations and WRF, to highlight the limits of the current WRF representation of the flow in complex terrain, and emphasize in turns the importance of the observational analysis you are presenting.
3. Section 3 needs references in several places in the part that was added in this revision.
4. P.12 l.6: the sentence starting with "However" is not clear. Same for the last sentence of the page, starting with "This".
5. P. 12 l.7: "in" instead of "is".
6. P. 19 l.3: some grammar errors in this sentence starting with "Secondly", please rephrase. Also, the following sentence is not clear: what do you mean by "same", what can you consider as sign of the wind speed vector? Please clarify.
7. P.19 l.6: 'exact opposite' instead of 'direct opposite'?
8. P. 19 l.8: please add commas to make the sentence easier to follow.

---

## Referee Report (RR2)

**Multi-lidar wind resource mapping in complex terrain**

by
Robert Menke et al.

**Review**

**General comments**

The revised version of the manuscript "Multi-lidar wind resource mapping in complex terrain" by Robert Menke et al. has been greatly improved. The description of the methods now follows a clearer structure and unnecessary parts were removed for more clarity. The intention of the authors to further qualify multi lidar measurements in complex terrain for wind resource assessment is now communicated more clear.

Nevertheless, some minor improvements can further strengthen the paper for more clarity.

1. The objective named in the introduction of the paper ("In this study, we present dual-Doppler lidar measurements and analyze flow structures in observed wind field for different atmospheric conditions") does not communicate the real intention of the paper (as stated e.g. in the title) to qualify multi lidar for wind resource assessment in complex terrain. I suggest rephrasing the second last paragraph in the introduction starting P2L13 to state this objective and the measures to achieve this goal (comparison of lidar data to anemometer data and flow model data).

2. The structure of the paper given at the end of the introduction is not consistent with the actual structure. Sections 2-4 contain methods as stated. Section 5 contains results and some discussions as stated, but with a misleading title ("Data analysis"). Section 6 contains discussions. Section 7 ("Conclusions") contains discussions and conclusions and is not listed in the introduction.

For a clearer structure I suggest to rename Section 5 to "Results" and to shift the few discussion parts from sections 5 and 7 to section 6. The content in section 7 should be limited to a very brief summary and real conclusions.

With these smaller but important changes and the few technical corrections (see below) done, I consider this nice work as ready for publication.

**Specific comments and technical corrections**

P2L23: Plural: wind fields

P2L23: Plural: ... compared to  WRF-LES simulations …

P3L12: I guess the type of Lidar is "Leosphere Windcube200S"

P3L20: … along the transect …

P3L20: in 80 m height above …

P3L26: below 0.5 % as  …

P6L20: Sentence hard to understand, please revise:  As no information, at the point of the model configuration, about the tree height was available, for the modeling domains, a randomly uniformly distributed forest height of 30 m ± 5 m was used.

P9L5: different name convention for Sls used here. Please correct.

P10L10: all four masts

P10L17: … wind speeds in the present study …

P15, caption Table 5: Introducing a second nomenclature for the met masts is not necessary and confuses the reader. Please stick to the original nomenclature.

---

## Author Response (AR2)

**Response to Referee 1**

Dear Anonymous Reviewer,

Thank you very much for reviewing the manuscript a second time and for your additional comments. We updated the manuscript according to your and the other reviewer's comments and suggestions.

Best regards,

The Authors

*Thank you for addressing most of my comments from the previous review. Please find below additional minor suggestions.*

*Note: references to page and lines in this list uses the marked-up version of the paper.*

1. *P.2 l.19: avoid repetition of "site".*

   The repetition of the work "site" has been removed from the sentence.

2. *P.2l.28: I still insist that it is important to emphasize the outcome of the comparison between observations and WRF, to highlight the limits of the current WRF representation of the flow in complex terrain, and emphasize in turns the importance of the observational analysis you are presenting.*

   We added the following sentence to the paragraph to already state the large deviations found in our comparison: "This comparison reveals large deviations of model data to the lidar measurements, which underlines the importance of in situ measurements in complex terrain."

3. *Section 3 needs references in several places in the part that was added in this revision.*

   We added the following references to the paragraph:

   Daniels, M. H., Lundquist, J. K., Mirocha, J. D., Wiersema, D. J., and Chow, F. K.: A new vertical grid nesting capability in the Weather Research and Forecasting (WRF) model, Mon. Wea. Rev., 144, 3725–3747, doi:10.1175/MWR-D-16-0049.1, 2016.

   Klemp, J. B., Dudhia, J., and Hassiotis, A. D.: An upper gravity-wave absorbing layer for NWP applications, Mon. Wea. Rev., 136, 3987–4004, doi:10.1175/2008MWR2596.1, 2008.

4. *P.12 l.6: the sentence starting with "However" is not clear. Same for the last sentence of the page, starting with "This".*

Thank you for pointing this out, both sentences have been reformulated and read now as follows:

"However, the measurements cannot be compared to studies that were purely designed for a comparison of different measurement technologies as e.g. done by Pauscher et al. (2016)."

"This affirms that the decisions, of using elevation angles below 5°, is adequate to measure the horizontal wind with the present scanning trajectories."

5. *P.12l.7: "in" instead of "is".*

Corrected.

6. *P. 19 l.3: some grammar errors in this sentence starting with "Secondly", please rephrase. Also, the following sentence is not clear: what do you mean by "same", what can you consider as sign of the wind speed vector? Please clarify.*

The grammar of the first sentence has been corrected. With "same" we refer to an identical magnitude of the speed up for opposite wind directions (to which were referred by the sign wind vector). The sentence has been reformulated for clarity: "The linear theory says that orography gives a speed up with the same magnitude for opposite wind directions."

7. *P.19l.6: 'exact opposite' instead of 'direct opposite'?*

Corrected.

8. *P.19l.8: please add commas to make the sentence easier to follow.*

Commas were added to sentence.

**Response to Referee 2**

Dear Anonymous Reviewer,

Thank you very much for reviewing the manuscript a second time and for your additional comments. We updated the manuscript according to your and the other reviewer's comments and suggestions.

Best regards,

The Authors

***General comments***
*The revised version of the manuscript "Multi-lidar wind resource mapping in complex terrain" by Robert Menke et al. has been greatly improved. The description of the methods now follows a clearer structure and unnecessary parts were removed for more clarity. The intention of the authors to further qualify multi lidar measurements in complex terrain for wind resource assessment is now communicated more clear. Nevertheless, some minor improvements can further strengthen the paper for more clarity.*
*1. The objective named in the introduction of the paper ("In this study, we present dual-Doppler lidar measurements and analyze flow structures in observed wind field for different atmospheric conditions") does not communicate the real intention of the paper (as stated e.g. in the title) to qualify multi lidar for wind resource assessment in complex terrain. I suggest rephrasing the second last paragraph in the introduction starting P2L13 to state this objective and the measures to achieve this goal (comparison of lidar data to anemometer data and flow model data).*

Thanks for this comment. We rephrased the paragraph to better point out our study's objectives, it reads now as follows: "In this study, we present dual-Doppler lidar measurements that aim to assess the wind resource along the ridges at Perdigão. The measurements are validated against mast based ultrasonic anemometers along the ridges and observed flow structures are analyzed. Moreover, the lidar measurements are compared to WRF-LES simulations with and without a parametrization of forest drag (Wagner et al., 2019a, b) to test the model capability in reproducing the observed flow structures. This comparison reveals large deviations of model data to the lidar measurements, which underlines the importance of in situ measurements in complex terrain."

*2. The structure of the paper given at the end of the introduction is not consistent with the actual structure. Sections 2-4 contain methods as stated. Section 5 contains results and some discussions as stated, but with a misleading title ("Data analysis"). Section 6 contains discussions. Section 7 ("Conclusions") contains discussions and conclusions and is not listed in the introduction.*
*For a clearer structure I suggest to rename Section 5 to "Results" and to shift the few discussion parts from sections 5 and 7 to section 6. The content in section 7 should be limited to a very brief summary and real conclusions.*

*With these smaller but important changes and the few technical corrections (see below) done, I consider this nice work as ready for publication.*

Thanks for pointing out that the description of the paper structure is not updated. We renamed section 5 as suggested and updated the paragraph in the introduction.

**Specific comments and technical corrections**
*P2L23: Plural: wind fields*

Corrected.

*P2L23: Plural: ... compared to a WRF-LES simulations ...*

Corrected.

*P3L12: I guess the type of Lidar is "Leosphere Windcube200S"*

Thanks for this remark, we corrected the manuscript.

*P3L20: ... along the transect ...*

Corrected.

*P3L20: in 80 m height above ...*

Corrected.

*P3L26: below 0.5 % as as ...*

Corrected.

*P6L20: Sentence hard to understand, please revise: As no information, at the point of the model configuration, about the tree height was available, for the modeling domains, a randomly uniformly distributed forest height of 30 m ± 5 m was used.*

The sentence has been restructured and reads now as follows: "As no information about the tree height at the point of the model configuration was available, a randomly uniformly distributed forest height of 30 m ± 5 m is used for the modeling domains."

*P9L5: different name convention for Sls used here. Please correct.*

Thanks for catching this mistake. The naming of the SLs has been corrcted.

*P10L10: all four masts*

Corrected.

*P10L17: ... wind speeds in the present study ...*

Corrected.

*P15, caption Table 5: Introducing a second nomenclature for the met masts is not necessary and confuses the reader. Please stick to the original nomenclature.*

The two naming conventions of the masts have been introduced during the measurement campaign and have been used both equally since then. We choose to show both conventions here to make it easier to cross reference the masts between publications for readers who are only familiar with one of the conventions.

**Multi-lidar wind resource mapping in complex terrain**

[revised manuscript text omitted]

25 analyzed. Moreover, the lidar measurements are compared to  WRF-LES simulations with and without a parametrization of forest drag (Wagner et al., 2019a, b) to test the model capability in reproducing the observed flow structures. This comparison reveals large deviations of model data to the lidar measurements, which underlines the importance of in situ measurements in complex terrain.

The paper is organized in the following way: Section 2 gives an overview of the Perdigão field campaign including a

30 description of lidar and mast measurements, Section 3 presents the WRF model setup. Section 4 introduces the applied data processing techniques.  Results are presented in Section 5, followed by  a discussion in Section 6. Conclusions are given in Section 7.

**2   Field campaign overview**

[revised manuscript text omitted]

**2.1.3 Scanning strategy**

The two trajectories, which follow the ridge top line 80 m AGL, were designed using the high precision terrain data. The traverses were 1.8 km long and described by points evenly spaced every 20 m. Accordingly we programmed the SLs to measure continuously along the trajectories by moving the beams through the trajectory points with the speed of 40 m s$^{-1}$ and an accumulation time of 500 ms. As a result, spatial averaging takes place normal to the beam direction. Along the beam, range gates were placed every 10 m, starting at 700 m, and extending to 2640 m (Table 1). Range gates  represent slices (a certain number of samples) of the sampled backscattered light which are analyzed to estimate the wind vector component along the LOS. Each range gate corresponds to a spatial  location for which the radial velocity is evaluated.  The SLs were configured to emit 200 ns laser pulses, and sample the resulting backscattered light with a frequency of 250 MHz. In the analysis of the back-scattered light each range gate is represented by 64 consecutive samples resulting in 
[revised manuscript text omitted]